# Transmission mode of watermelon silver mottle virus by *Thrips palmi*

**De-Fen Mou**[1¤]**, Wei-Te Chen**[1]**, Wei-Hua Li**[1]**, Tsung-Chi Chen**[2]**, Chien-Hao Tseng**[1]**, Li-Hsin Huang**[3]**, Jui-Chu Peng**[4]**, Shyi-Dong Yeh**[5]**, Chi-Wei Tsai**[1]*

**1** Department of Entomology, National Taiwan University, Taipei, Taiwan, **2** Department of Medical Laboratory Science and Biotechnology, Asia University, Taichung, Taiwan, **3** Pesticide Application Division, Taiwan Agricultural Chemicals and Toxic Substances Research Institute, Taichung, Taiwan, **4** Division of Crop Environment, Tainan District Agricultural Research and Extension Station, Tainan, Taiwan, **5** Department of Plant Pathology, National Chung Hsing University, Taichung, Taiwan

¤ Current address: Fort Lauderdale Research and Education Center, Department of Entomology and Nematology, University of Florida, Fort Lauderdale, Florida, United States of America
* chiwei@ntu.edu.tw

**Data Availability Statement:** All relevant data are within the manuscript and its Supporting information files.

**Funding:** This research was supported by the Ministry of Science and Technology of Taiwan

## Abstract

Thrips and thrips-transmitted tospoviruses cause significant losses in crop yields worldwide. The melon thrips (*Thrips palmi*) is not only a pest of cucurbit crops, but also a vector that transmits tospoviruses, such as the watermelon silver mottle virus (WSMoV). Vector transmission of tospoviruses has been well studied in the tomato spotted wilt virus (TSWV)–*Frankliniella occidentalis* model system; however, until now the transmission mode of WSMoV by *T. palmi* has not been sufficiently examined. The results of the transmission assays suggest that *T. palmi* transmits WSMoV in a persistent manner, and that the virus is mainly transmitted by adults, having been ingested at the first-instar larval stage. Complementary RNAs corresponding to the *NSm* and *NSs* genes of WSMoV were detected in viruliferous thrips by reverse transcription-polymerase chain reaction; NSs protein was also detected in viruliferous thrips by western blotting, verifying the replication of WSMoV in *T. palmi*. Furthermore, we demonstrated that in thrips infected with WSMoV at the first-instar larval stage, the virus eventually infected various tissues of the adult thrips, including the primary salivary glands. Taken together, these results suggest that *T. palmi* transmits WSMoV in a persistent-propagative mode. The results of this study make a significant contribution to the understanding of the transmission biology of tospoviruses in general.

## Introduction

Thrips and thrips-transmitted tospoviruses cause significant yield losses in both food crops and ornamentals in many countries across the globe [1, 2]. Tospovirus-incited diseases have become a major constraint on the production of vegetables, fruits, and legumes worldwide. Of the tospoviruses, tomato spotted wilt virus (TSWV) is the most devastating due to its extremely wide host range [3]. TSWV is transmitted from plant to plant by several thrips species, with the western flower thrips (*Frankliniella occidentalis*) being its most efficient vector [4].

(grant number NSC 100-2321-B-002-035-MY3). The funders had no role in study design, data collection and analysis, decision to publish, or preparation of the manuscript.

**Competing interests:** The authors have declared that no competing interests exist.

Watermelon silver mottle virus (WSMoV) is another tospovirus which affects cucurbit production in Asia, resulting in yield reduction and unmarketable fruits [5, 6]. The melon thrips (*Thrips palmi*) is the only identified vector of WSMoV [6], whilst the vector competence of other thrips species has not been investigated.

Tospoviruses belong to the genus *Orthotospovirus* of the family *Tospoviridae* [7]. To date, the International Committee on Taxonomy of Viruses lists 18 tospoviruses as recognized species. TSWV, impatiens necrotic spot virus (INSV), and iris yellow spot virus (IYSV) occur in five continents of the world, whereas WSMoV has only been recorded in Asian countries [3]. TSWV and INSV have wide host ranges, comprising more than 900 and 300 plant species, respectively [2, 3]. WSMoV only infects cucurbit species and some experimental hosts [5, 6].

In the field, tospoviruses are transmitted exclusively by thrips [8]. Mound (2002) [9] suggested a coevolution for the transmission specificity between tospoviruses and their vector thrips. While there are over 7,700 thrips species identified, so far only 18 species have been reported to transmit tospoviruses [4, 8, 10, 11]. All vector thrips belong to the family Thripidae and subfamily Thripinae [4], with the exception of *Neohydatothrips variabilis* which belongs to subfamily Sericothripinae [12]. *Frankliniella occidentalis* and *T. palmi* have the ability to transmit at least seven tospovirus species each, with no overlap [4, 8]. TSWV is transmitted by several thrips species, including *F. occidentalis* [13–18], whereas the only known vector of WSMoV is *T. palmi* [6]. Tospoviruses are transmitted by mechanical inoculation with varying degrees of difficulty [19], and none are seed-transmitted [5, 20, 21].

Insect transmission of plant viruses is categorized into four transmission modes: non-persistent, semi-persistent, persistent-circulative, and persistent-propagative [22, 23]. The characteristics used to distinguish each mode are: acquisition access period (AAP), inoculation access period (IAP), latent period, retention time in vector, presence in vector's hemolymph, and multiplication within the vector [22, 23]. Thrips transmission of tospoviruses includes several days of latent period, followed by lifelong retention of the virus, with the viruses replicating within various tissues of the vector [8].

Our current understanding of tospovirus transmission is largely from studies on the TSWV–*F. occidentalis* model system. *Frankliniella occidentalis* transmits TSWV in a persistent-propagative mode [8, 23]. To sum up, first- and second-instar larvae feeding on a TSWV-infected plant can acquire the virus which is subsequently transmitted during the adult stage or, rarely, during the second-instar stage. Uniquely, whilst adult thrips can acquire the virus by feeding, they cannot transmit the virus that is acquired at the adult stage [24]. During the course of vector transmission, TSWV infects various tissues and replicates in the cells of *F. occidentalis* [25–28]. Once *F. occidentalis* has acquired TSWV, it persistently transmits the virus for the rest of its life; however, TSWV is not transmitted to the progeny of its vector thrips [29].

Researchers may consider that the transmission biology of other tospoviruses should be the same as that of the well-studied TSWV model. Therefore, to examine the competence of a thrips species to act as a vector of a given tospovirus, most studies assumed virus acquisition during the larval stage and transmission during the adult stage (e.g., [10, 11, 14, 16–18, 30, 31]). However, it is possible that the transmission biology of other tospoviruses differs from that of TSWV. In fact, the possibility of non-persistent and semi-persistent transmission modes have been rarely studied.

Based on a phylogenetic analysis of nucleoprotein amino acid sequences, WSMoV belongs to the Asian group, while TSWV is a representative of the American group [3, 32], implying that WSMoV is distantly related to TSWV. Our objective was to examine the transmission mode of WSMoV, a virus that is distantly related to TSWV, which is considered as a well-

studied prototype. The results of this study will broaden our understanding of the transmission biology of tospoviruses in general.

## Materials and methods

### Insect, virus, and plants

The sources of the *T. palmi* colony and WSMoV Tainan isolate were previously described in Chen *et al.* [33]. The thrips colony was maintained on bean (*Phaseolus coccineus*) seedlings in 2-L glass beakers sealed with fine fabric (150 mesh/inch, pore size 160 μm) in an environmental chamber at 25°C, 70% relative humidity (RH), and a photoperiod of L:D 16:8 h. Adult females were transferred to cucumbers (*Cucumis sativus*) using a fine brush, and pollen was provided to enhance oviposition. Newly hatched first- and second-instar larvae were collected from the surface of the cucumbers using a fine brush.

WSMoV was maintained in watermelon (*Citrullus lanatus* cv. Empire no. 2, Known-You Seed Co., Taiwan) seedlings by vector transmission using *T. palmi*. WSMoV-infected source plants and non-infected watermelon seedlings were prepared and maintained as previously described [33]. Watermelon and bean seedlings used in this study were grown from seeds and cultured in an environmental chamber at 25°C, 70% RH, and a photoperiod of L:D 16:8 h. All experiments had negative controls, which were from the same batch of watermelon seedlings but had not been exposed to thrips. These negative controls were kept with the test plants inoculated with WSMoV by viruliferous *T. palmi* within the same environmental chamber until the reverse transcription-polymerase chain reaction (RT-PCR) assay was conducted (see below). None of the control plants tested positive for WSMoV.

### Non-persistent transmission mode

The transmission assays for the non-persistent transmission mode were conducted with three treatments using different developmental stages of the thrips: 1) virus acquisition and transmission by first-instar larvae, 2) virus acquisition and transmission by second-instar larvae, and 3) virus acquisition and transmission by adults. For the test insects, we used first-instar larvae within 4 h of hatching, second-instar larvae within 4 h of molting, and adults within 48 h of emergence. Watermelon seedlings at the one-true-leaf stage were used as healthy test plants. After a 1-h starvation period, the thrips were given an acquisition access period (AAP) of 15 min to acquire the virus from detached WSMoV-infected watermelon leaves in plastic Petri dishes (9 cm dia.). Subsequently, five potentially viruliferous thrips were transferred to each test plant enclosed in a 1-L beaker sealed with a fine fabric (150 mesh/inch) for a 2-h inoculation access period (IAP). Ten replicate test plants for each treatment were inoculated with their respective thrips. After the IAP, the inoculated test plants were sprayed with a systemic insecticide (acetamiprid) to kill the thrips. We maintained the thrips-inoculated test plants within an environmental chamber at the aforementioned conditions for disease development until the RT-PCR assay (see below). The experiment was repeated three times.

### Semi-persistent transmission mode

Similar experimental protocols to those used for the non-persistent transmission mode were conducted to investigate whether *T. palmi* transmits WSMoV semi-persistently. The transmission assays consisted of three treatments: 1) virus acquisition and transmission by first-instar larvae, 2) virus acquisition and transmission by second-instar larvae, and 3) virus acquisition and transmission by adults. The test thrips were given an AAP of 2 h on detached WSMoV-infected watermelon leaves in Petri dishes (9 cm dia.) to acquire the virus, and then potentially

viruliferous thrips were allowed to feed on test plants at a density of five thrips per plant for a 24-h IAP. Ten replicate test plants were used per treatment. After the IAP, the inoculated test plants were sprayed with acetamiprid and maintained using the aforementioned protocol. The experiment was repeated three times.

## Persistent transmission mode

Similar experimental protocols to those used for non-persistent transmission mode were conducted to investigate whether *T. palmi* transmits WSMoV persistently. The transmission assays consisted of four treatments: 1) virus acquisition by first-instar larvae and transmission by adults, 2) virus acquisition by first-instar larvae and transmission by second-instar larvae, 3) virus acquisition by second-instar larvae and transmission by adults, and 4) virus acquisition and transmission by adults. The test thrips were given an AAP of 48 h on detached WSMoV-infected watermelon leaves in Petri dishes (9 cm dia.) to acquire the virus. After the AAP, first- and second-instar larvae were individually reared on a piece of bean leaf ($1.5 \times 1.5$ cm) enclosed in a glass vial ($4.5$ cm $\times 1.5$ cm dia.) sealed with parafilm and incubated at 25°C until they developed to second-instar larvae or adults through various latent periods. The adults were sexed after emergence. Subsequently, one or five potentially viruliferous thrips were transferred to each test plant for a 48-h IAP. One thrips per test plant for virus inoculation was used to compare the transmission efficiencies of adult females and males for treatment 1), and five thrips per test plant were used to account for the possible low transmission efficiencies for the other treatments. Ten replicate test plants were used per treatment. After the IAP, the inoculated test plants were sprayed with acetamiprid and maintained using the aforementioned protocol. The experiment was repeated three times.

## Virus detection by one-step RT-PCR

WSMoV-infected leaves were subjected to RNA extraction and one-step RT-PCR analysis after the end of the AAP to confirm infection with the virus. All virus-infected source leaves tested positive for WSMoV. All inoculated test plants were examined using one-step RT-PCR two weeks after the end of the IAP. RNA extraction and one-step RT-PCR analysis were conducted as previously described [33]. The vector transmission rate was calculated as the percentage of test plants infected with WSMoV.

## Virus-infected tissues of *T. palmi*

To examine the WSMoV-infected tissues and organs of *T. palmi*, an indirect immunofluorescence assay was performed as previously described by Montero-Astúa *et al.* [34]. First-instar larvae of *T. palmi* were allowed to feed on WSMoV-infected watermelon leaves for 48 h and then transferred to bean seedlings until they developed into adults. Potentially viruliferous adult females were collected for the immunofluorescence assay. Tissues and organs of the thrips were dissected in ice-cold 0.01 M phosphate-buffered saline (PBS, pH 7.4) under a stereomicroscope. The dissected tissues were fixed with 4% paraformaldehyde for 1 h in a humid box at room temperature. After fixation, the tissues were rinsed twice with PBS containing 0.1% Triton-X 100 (PBST), and then incubated with PBST overnight at 4°C. Subsequently, the tissues were rinsed with PBS and blocked with 5% bovine serum albumin (BSA) in PBS for 30 min at room temperature to prevent non-specific binding of the antibody. Afterwards, the tissues were rinsed three times with PBST and then incubated with a polyclonal rabbit antiserum against WSMoV nucleocapsid protein (200× dilution in PBST with 0.1% BSA) for 1.5 h at room temperature. The tissues were rinsed three times with PBST and then incubated with a goat anti-rabbit immunoglobulin G (IgG) antiserum conjugated with Alexa Fluor 555 (200×

dilution, Invitrogen) for 2 h at room temperature. After rinsing three times with PBST, the tissues were incubated with phalloidin conjugated with Alexa Fluor 633 (40× dilution, Invitrogen) for 2 h at room temperature. Finally, the specimens were rinsed three times with PBST to remove any non-specific binding and mounted with SlowFade Gold antifade reagent with DAPI (Invitrogen). Slides were kept at 4°C until examined with a Leica TCS SP5 II confocal laser-scanning microscope the following day.

## Detection of viral complementary RNA by two-step RT-PCR

Two-step RT-PCR was used to detect the viral complementary RNA (vcRNA) corresponding to the *NSm* and *NSs* genes in viruliferous thrips and infected plants. vcRNA is first replicated from viral genomic RNA (vgRNA), and then progeny vgRNA is generated from the vcRNA [35]. The presence of vcRNA implies the replication of WSMoV in its vector. Total RNA was extracted from single thrips or watermelon leaf tissue using TRIzol Reagent (Invitrogen) according to the manufacturer's instructions. Total RNA of non-viruliferous thrips and WSMoV-infected watermelon leaf tissue served as negative and positive controls, respectively. vcRNA in total RNA samples was reverse-transcribed to cDNA using a GoScript Reverse Transcription System (Promega) according to the manufacturer's instructions. We designed two primers: NSm_372 (5'– ATGACTCTCTTGTTGGTAATGG –3') and NSs_73 (5'– ACTGCAAAGAATGCTGCTTC –3') that would specifically anneal to the complementary strands of the WSMoV M and S segments, respectively. cDNA was amplified by PCR using NSm_372 and NSm_668 (5'– GTTGCATGCACTGCTTAGG –3') as primers for the WSMoV M segment and NSs_73 and NSs_378 (5'– CTTCACACCTGGTTTCCTTAC –3') for the WSMoV S segment. PCR conditions were as follows: 95°C for 5 min, followed by 35 cycles of 95°C for 30 sec, 52°C for 30 sec, and 72°C for 30 sec, with a final extension step at 72°C for 5 min. Expected sizes of the PCR products were 297 bp and 306 bp for partial *NSm* and *NSs* genes, respectively. The PCR products were purified using a QIAquick PCR purification kit (Qiagen) and then directly sequenced in both directions by Sanger sequencing. The identities of the sequences were confirmed by conducting BLAST searches in NCBI GenBank using the BLASTn algorithm. The experiment was repeated three times.

## Western blot analysis

Western blotting was used to detect WSMoV NSs protein in viruliferous thrips and infected plants. Crude antigen was extracted from pools of 20 thrips or watermelon leaf tissue using Laemmli sample buffer (Bio-Rad) according to the manufacturer's instructions. Crude extracts from non-viruliferous thrips and WSMoV-infected watermelon leaves were served as negative and positive controls, respectively. Female and male thrips were treated separately. The proteins were separated from others by sodium dodecyl sulfate–polyacrylamide gel electrophoresis (SDS-PAGE), transferred to a polyvinylidene fluoride (PVDF) membrane, and labelled by primary and secondary antibodies as previously described by Mahmood and Yang [36]. A monoclonal mouse antibody against the NSs protein (5,000× dilution) and a goat anti-mouse IgG antiserum conjugated with horseradish peroxidase (25,000× dilution; Invitrogen) were used in this study. The target bands were detected with LumiFlash Femto Chemiluminescent substrate (Visual Protein, Taiwan). The experiment was repeated three times.

## Statistical analyses

For the transmission assays, data from the three replicates were pooled for further statistical analysis because there were no significant differences of variance among them (S1 Table). Transmission rates among treatments in each transmission mode were compared using $\chi^2$

test. Swallow estimator [37] was used to estimate the probability that a single thrips transmits the virus when groups of thrips were used for virus inoculation. All data analyses were performed using SPSS Statistics 22.0.

## Results

### Non-persistent transmission mode

To assess whether *T. palmi* transmits WSMoV non-persistently, first- and second-instar larvae and adults were given characteristic short AAP and IAP without a latent period [22]. The results of the transmission assays revealed that *T. palmi* could not transmit WSMoV with a 15-min AAP and a 2-h IAP, irrespective of the developmental stage of the thrips used for virus acquisition and inoculation (Table 1). The transmission rates of the three developmental stages of the thrips were 0%. WSMoV-infected source leaves were subjected to RT-PCR assay after the end of the AAP, and all tested positive for WSMoV. The results suggest that *T. palmi* cannot transmit WSMoV in a non-persistent manner.

### Semi-persistent transmission mode

To assess whether *T. palmi* transmits WSMoV semi-persistently, first- and second-instar larvae and adults were given the characteristic AAP and IAP of semi-persistent transmission without a latent period [22]. The results of the transmission assays revealed that first- and second-instar larvae transmitted WSMoV with a 2-h AAP and a 24-h IAP, although the transmission rates were very low (Table 1). Adults could not transmit WSMoV under semi-persistent conditions (Table 1). There were no significant differences of variance among data from three replicates ($\chi^2$ test, $P > 0.05$; S1 Table), so the data were pooled for further statistical analysis. The transmission rates of first- and second-instar larvae were 3.3% and 6.7%, respectively, and they were not significantly different ($\chi^2 = 0.35$; $df = 1$; $P = 0.55$). The Swallow estimator (Ps) was about 1% for both first- and second-instar larvae (Table 1). Virus-infected source leaves were subjected to RT-PCR assay after the end of the AAP, and all tested positive for WSMoV. The results suggest that only about 1% of *T. palmi* larvae transmit WSMoV in a semi-persistent manner.

### Persistent transmission mode

To assess whether *T. palmi* transmits WSMoV persistently, first- and second-instar larvae and adults were given a 48-h AAP, various latent periods, and a 48-h IAP successively. The results showed that there were no significant differences of variance among data from three replicates ($\chi^2$ test, $P > 0.05$; S1 Table), so the data were pooled for further statistical analysis. The results of the transmission assays revealed that WSMoV transmission occurred in all four treatments (Table 1), and they were significantly different ($\chi^2 = 67.43$; $df = 3$; $P = 0.0001$). Virus acquisition by first-instar larvae and transmission by adults yielded the highest transmission efficiency (76.7%). This was the treatment with the longest latent period (about 10 d). The transmission efficiencies of adult females and males were 71.4% and 81.3%, respectively, and there was no significant difference between them ($\chi^2 = 0.40$; $df = 1$; $P = 0.53$). When the virus was acquired by first-instar larvae and transmitted by second-instar larvae, the transmission efficiency was 3.3% (Ps = 0.7%). Adult thrips transmitted WSMoV with low efficiency when the virus was acquired at the second-instar larval or adult stages (3.3% and 6.7%, respectively), and they were not significantly different ($\chi^2 = 0.35$; $df = 1$; $P = 0.55$). Virus-infected source leaves were subjected to RT-PCR assay after the end of the AAP, and all tested positive for

**Table 1. Transmission mode and transmission efficiency of watermelon silver mottle virus by *Thrips palmi*.**

| Transmission mode | Virus-acquiring instar | Virus-inoculating instar | N | Transmission efficiency[a] | Ps[b] |
|---|---|---|---|---|---|
| *Non-persistent transmission* | | | | | |
| | First-instar larva | First-instar larva | 30 | 0% | 0% |
| | Second-instar larva | Second-instar larva | 30 | 0% | 0% |
| | Adult | Adult | 30 | 0% | 0% |
| *Semi-persistent transmission* | | | | | |
| | First-instar larva | First-instar larva | 30 | 3.3% | 0.7% |
| | Second-instar larva | Second-instar larva | 30 | 6.7% | 1.4% |
| | Adult | Adult | 30 | 0% | 0% |
| *Persistent transmission* | | | | | |
| | First-instar larva | Adult | 30 | 76.7% | – |
| | First-instar larva | Second-instar larva | 30 | 3.3% | 0.7% |
| | Second-instar larva | Adult | 30 | 3.3% | 0.7% |
| | Adult | Adult | 30 | 6.7% | 1.4% |

[a] The transmission efficiency was calculated as the percentage of test plants positive for WSMoV examined by RT-PCR assays.

[b] Ps: Swallow estimator, probability that a single thrips transmits the virus. $Ps = 1-(R/N)^{1/k}$ where k = number of thrips per test plant; N = number of test plants; R = number of negative plants.

WSMoV. The results suggest that *T. palmi* transmits WSMoV in a persistent manner and the virus is mainly transmitted by adults, having being ingested during the first-instar larval stage.

## Virus infection in *T. palmi*

Organs of *T. palmi* were dissected and examined with an immunofluorescence assay to verify WSMoV infection within various tissues (Figs 1 and 2). The confocal images of non-viruliferous and WSMoV-infected *T. palmi* are representative images from 12 non-viruliferous and 19 viruliferous thrips examined in this study. The alimentary tract of *T. palmi* is divided into the foregut, midgut, and hindgut, with the midgut is further divided into midgut 1, midgut 2, and midgut 3 (Fig 1). No Alexa Fluor 555 signal was detected in the organs of non-viruliferous thrips (Fig 1). The fluorescence signal of Alexa Fluor 555, indicating WSMoV infection, was detected in the primary salivary glands, foregut, midgut, hindgut and Malpighian tubules (Fig 2A–2D), but not in the ovaries (Fig 2E) of viruliferous thrips. The results demonstrate that WSMoV infects the primary salivary glands, foregut, midgut, hindgut and Malpighian tubules of *T. palmi* adults that acquired the virus during the first-instar larval stage.

## Virus replication in *T. palmi*

To confirm the replication of WSMoV in viruliferous thrips, the vcRNA corresponding to the *NSm* gene (on the M segment) and the *NSs* gene (on the S segment) of WSMoV were detected by two-step RT-PCR. Fragments of the *NSm* and *NSs* genes were amplified from the total RNA extracted from viruliferous thrips and WSMoV-infected watermelon leaves (Fig 3). Identities of the PCR products were confirmed by DNA sequencing and BLASTn searches. The sequences of partial *NSm* and *NSs* genes were 100% identical to WSMoV isolate DD6 the M and S segments in GenBank (accession numbers DQ157768 and AY864852, respectively). The presence of the vcRNA of *NSm* and *NSs* genes in viruliferous thrips implies the replication of WSMoV in viruliferous *T. palmi*.

To further confirm the replication of WSMoV in viruliferous thrips, WSMoV NSs protein was detected by western blotting in female and male thrips. A protein with a molecular size of

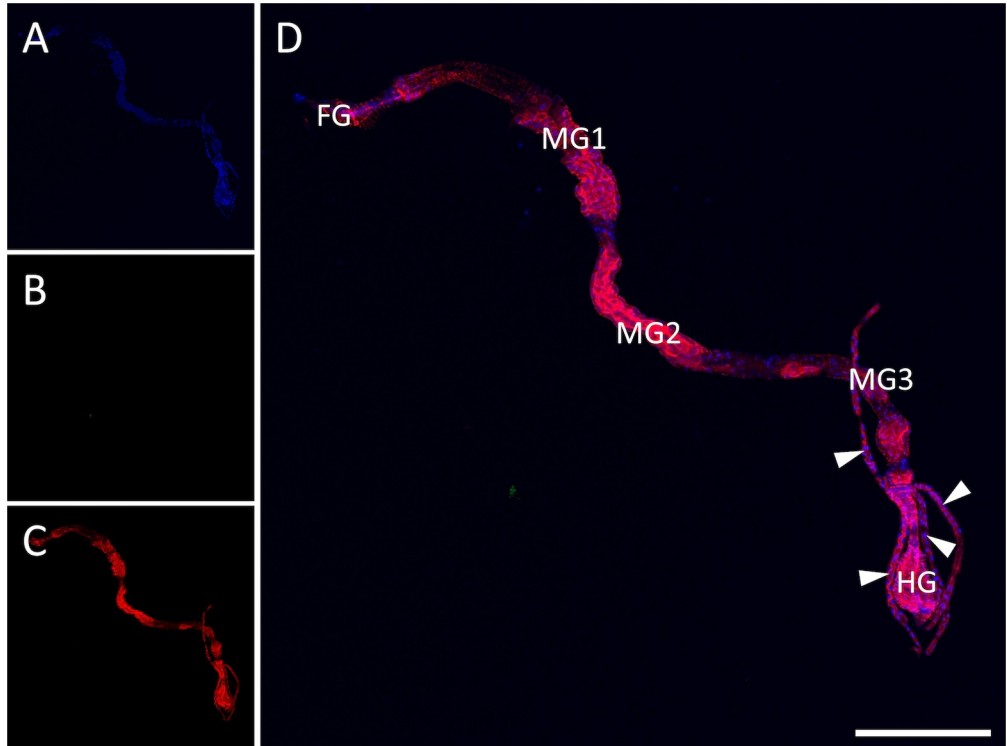

**Fig 1. Indirect immunofluorescence detection of watermelon silver mottle virus (WSMoV) in non-viruliferous *Thrips palmi*.** (A) Nuclei of cells stained with the DNA selective dye DAPI. (B) No WSMoV antigen detected. (C) Actin stained with phalloidin conjugated with Alexa Fluor 633. (D) Merged image, indicating that no WSMoV antigen was detected in the alimentary tract. WSMoV antigen was detected using rabbit antiserum against WSMoV nucleocapsid protein and a secondary antibody conjugated with Alexa Fluor 555. Actin was stained to delineate cell boundaries and tissue types. The confocal image depicting alimentary tract of a thrips is a representative image from 12 non-viruliferous thrips examined in this study. FG: foregut, MG: midgut, HG: hindgut. Arrowheads point to Malpighian tubules. Scale bar = 250 μm.

about 50 kDa was identified in the protein extract of viruliferous thrips and WSMoV-infected watermelon leaves (Fig 4). The presence of the NSs protein in viruliferous thrips confirms the replication of WSMoV in viruliferous *T. palmi*.

## Discussion

To examine the competence of a thrips species to act as a vector of a given tospovirus, most studies have suggested that the thrips must acquire the virus at the larval stage and transmit it at the adult stage. For example, TSWV is ingested by first-instar larvae and transmitted by adults of *F. occidentalis*, *F. schultzei*, *F. intonsa*, *T. tabaci* [14], *F. fusca* [13], *F. bispinosa* [17], and *F. cephalica* [18]. Similarly, INSV is ingested by first-instar larvae and transmitted by adults of *F. occidentalis* and *F. intonsa* [14, 31]. *Thrips palmi* is the only identified vector of WSMoV [6]. Because Chen *et al.* [38] proposed that *T. palmi* transmits WSMoV both non-persistently and persistently, we reexamined the transmission mode of WSMoV. In this study, we conducted a series of experiments to examine whether *T. palmi* transmits WSMoV non-persistently, semi-persistently, or persistently.

*Thrips palmi* most efficiently transmitted WSMoV with an AAP of 48 h at the first-instar larval stage followed by a 10-d latent period until emergence of adults, and an IAP of 48 h at

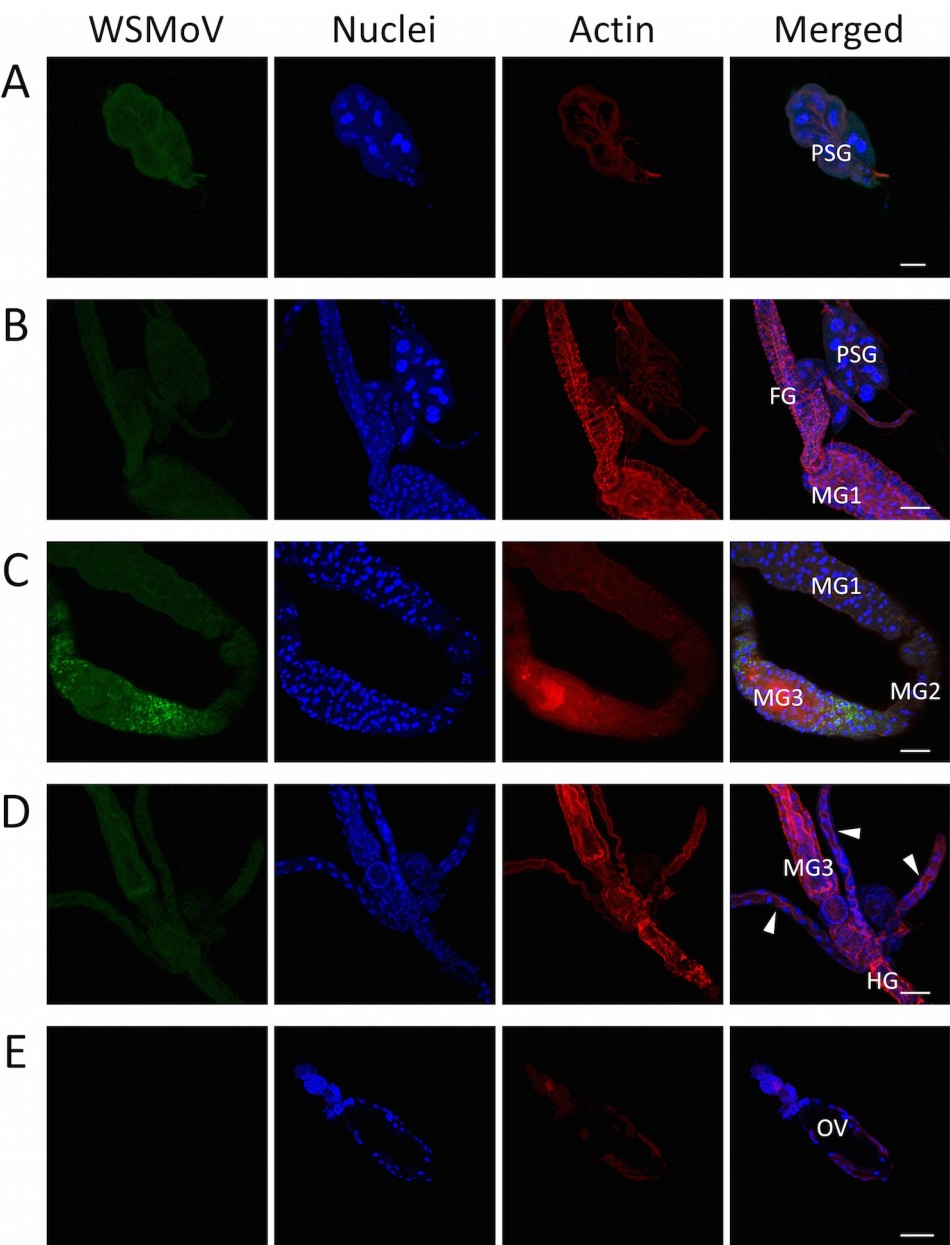

**Fig 2. Indirect immunofluorescence detection of WSMoV in viruliferous *T. palmi*.** WSMoV antigen was detected using rabbit antiserum against WSMoV nucleocapsid protein and a secondary antibody conjugated with Alexa Fluor 555. The DNA selective dye DAPI was used to stain the cells' nuclei. Actin was stained with phalloidin conjugated with Alexa Fluor 633. (A), (B), (C), (D), and (E) are organs of five different thrips, and a total of 19 viruliferous thrips were examined in this study. PSG: primary salivary gland, FG: foregut, MG: midgut, HG: hindgut, OV: ovariole. Arrowheads point to Malpighian tubules. Scale bars = 50 μm.

the adult stage (Table 1). WSMoV was not transmitted by *T. palmi* non-persistently, and the probability of a single thrips transmitting the virus was less than 1.4% for semi-persistent transmission (Table 1). The transmission efficiency of adult thrips is influenced by the larval stage at which the tospovirus is acquired from infected plant tissues [8]. The "developmental-stage dependent virus acquisition" of WSMoV transmission by *T. palmi* is in line with other

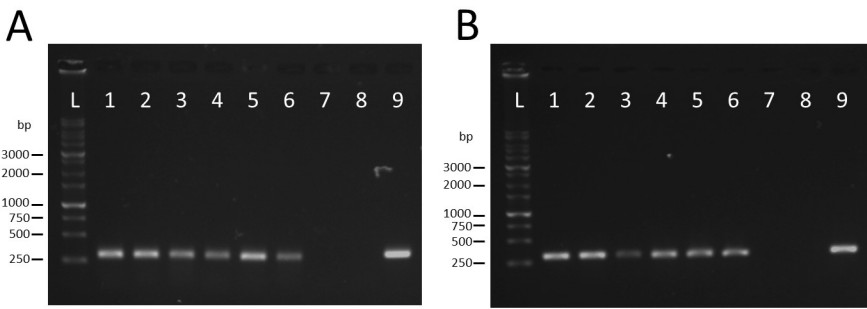

**Fig 3. Agarose gel electrophoresis analysis of viral complementary RNA (vcRNA) corresponding to (A) *NSm* and (B) *NSs* genes of WSMoV in viruliferous *T. palmi*.** vcRNA was reverse-transcribed to cDNA, and then the cDNA was amplified by PCR. Lanes: L, 1 kb DNA ladder; 1–3, viruliferous females; 4–6, viruliferous males; 7, non-viruliferous female; 8, non-viruliferous male; 9, WSMoV-infected watermelon leaf.

tospoviruses transmitted by vector thrips. Transmission of capsicum chlorosis virus (CaCV) and TSWV by thrips is also developmental-stage dependent [24, 39].

According to the characteristics of the different transmission modes [23], our data demonstrate that *T. palmi* transmits WSMoV in a persistent-propagative mode. When *T. palmi* acquired WSMoV at the first-instar larval stage, the virus eventually infected various tissues of the adult thrips, including the salivary glands (Fig 2). This is consistent with results showing that *T. palmi* transmitted WSMoV at the adult stage. The detection of vcRNA and NSs protein in viruliferous *T. palmi* (Figs 3 and 4) also implies the replication of WSMoV in the vector thrips. This knowledge is consistent with previous studies on TSWV, which was found to be transmitted by several thrips species in a persistent-propagative mode [8].

Our data demonstrate that when WSMoV is acquired at the first-instar larval stage, it is accumulated in the foregut, midgut, hindgut, Malpighian tubules, and salivary glands of *T. palmi* adults (Fig 2). Previous histological studies, mainly focused on TSWV transmitted by other thrips, have described TSWV infecting the foregut, midgut, Malpighian tubules, and salivary glands in *T. setosus*, *F. occidentalis*, and *T. tabaci* [15, 27, 28, 40]. The dissemination dynamics of TSWV is very similar in these three thrips species. TSWV infects the proximal midgut region, second and third midgut regions, and foregut sequentially, and lastly infects the salivary glands [15, 27, 28]. Wijkamp *et al.* [41] demonstrated that TSWV replicates in the salivary glands of *F. occidentalis*; the virus particles mature in the salivary vesicles and then are

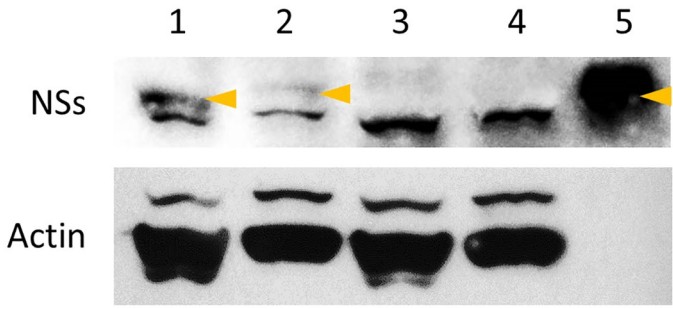

**Fig 4. Western blot analysis of WSMoV NSs protein in viruliferous *T. palmi*.** NSs protein with a molecular size of about 50 kDa was detected from the protein extract of viruliferous thrips. Western blotting of Actin in *T. palmi* was conducted as loading control. Lanes: 1, 20 viruliferous females; 2, 20 viruliferous males; 3, 20 non-viruliferous females; 4, 20 non-viruliferous males; 5, WSMoV-infected watermelon leaf. Arrowheads point to NSs protein.

transported to the salivary ducts. Competent thrips species support the replication of tospo-viruses and eject virus particles into the plant through their saliva.

In this study, we also compared sexual differences in the transmission efficiency of WSMoV by *T. palmi*. There was no significant difference in the transmission efficiency between adult females and males. Similar results have been reported in the transmission of INSV by *F. occidentalis* [31], CaCV by *Ceratothripoides claratris* [39] and Hippeastrum chlorotic ringspot virus by *Taeniothrips eucharii* [11]. In contrast, the transmission efficiency of TSWV varies between females and males of *F. occidentalis* [42–45], *F. cephalica* [18], and *T. tabaci* [46]. A sexual difference in transmission efficiency was also reported in the transmission of INSV by *F. intonsa* [31]. In the aforementioned cases, adult males transmit tospovirus more efficiently than adult females. Males of *F. occidentalis* have higher mobility but feed less frequently and intensively than females; thus, males cause less damage to the plant cells, but make more effective inoculation punctures than females [43]. Furthermore, males of *F. occidentalis* infected with TSWV feed more frequently and perform more non-ingestion probes than uninfected males [47]. This may explain the different transmission efficiencies of TSWV between the sexes in *F. occidentalis*; however, it is unclear whether this is the case in other thrips species. Studies on the activity and feeding behavior of females and males of *T. palmi* may help to address this question.

Our results clarify that *T. palmi* transmits WSMoV in a persistent-propagative mode, and not in a non-persistent or semi-persistent manner. The virus is mainly transmitted by adult thrips, after being ingested in the first-instar larval stage. TSWV was proven to be transmitted by several thrips species in a persistent-propagative mode [8]. Although most tospoviruses have not been tested for non-persistent or semi-persistent transmission modes, they are confirmed as persistently transmitted viruses with their corresponding vector thrips. Since WSMoV and TSWV are not phylogenetically closely related, and neither are their vectors *T. palmi* and *F. occidentalis*, there is a high probability that all tospoviruses are transmitted in a persistent-propagative mode and have "developmental-stage dependent virus acquisition" for their vector transmission. It also fits the hypothesis that the mode of plant virus transmission is a stable evolutionary trait that is always the same for a given virus genus [48]. The results of this study make a significant contribution to understanding of the transmission biology of tospoviruses in general.

## Supporting information

**S1 Table. Transmission mode and transmission rate of watermelon silver mottle virus by *Thrips palmi*.**
(PDF)

**S1 Raw images.**
(PDF)

## Acknowledgments

The authors would like to thank Alexander C. Barton and Reina L. Tong for editing this manuscript. We also thank the Joint Center for Instruments and Researches, College of Bioresources and Agriculture, National Taiwan University for equipment support of the confocal microscope.

## Author Contributions

**Conceptualization:** De-Fen Mou, Li-Hsin Huang, Jui-Chu Peng, Chi-Wei Tsai.

**Formal analysis:** De-Fen Mou.

**Investigation:** De-Fen Mou, Wei-Te Chen, Wei-Hua Li, Chien-Hao Tseng.

**Methodology:** Tsung-Chi Chen, Shyi-Dong Yeh.

**Resources:** Tsung-Chi Chen, Li-Hsin Huang, Jui-Chu Peng, Shyi-Dong Yeh.

**Supervision:** Chi-Wei Tsai.

**Validation:** Wei-Hua Li.

**Writing – original draft:** De-Fen Mou, Wei-Te Chen, Chi-Wei Tsai.

**Writing – review & editing:** De-Fen Mou, Chi-Wei Tsai.

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
