## [Decision Letter · Decision Letter 0]

12 Aug 2020

PONE-D-20-21623

Transmission mode of watermelon silver mottle virus by Thrips palmi

PLOS ONE

Dear Dr. Tsai,

Thank you for submitting your manuscript to PLOS ONE. After careful consideration, we feel that it has merit but does not fully meet PLOS ONE’s publication criteria as it currently stands. Therefore, we invite you to submit a revised version of the manuscript that addresses the points raised during the review process.

The manuscript has been evaluated by two external reviewers. Please, see that Reviewer #2 raised major concerns regarding to experiments, data analyses and conclusions drawn in the manuscript and rejected its publication. However, considering the scientific contribution of this manuscript, we would like to offer you a chance to resolve the issues raised by this reviewer and we recommend you to make a major revision of the manuscript.

We look forward to receiving your revised manuscript.

Kind regards,

Livia Maria Silva Ataide

Academic Editor

PLOS ONE

Journal Requirements:

Reviewers' comments:

Reviewer's Responses to Questions

**Comments to the Author**

1. Is the manuscript technically sound, and do the data support the conclusions?

Reviewer #1: Yes

Reviewer #2: Partly

2. Has the statistical analysis been performed appropriately and rigorously? 

Reviewer #1: No

Reviewer #2: No

3. Have the authors made all data underlying the findings in their manuscript fully available?

Reviewer #1: Yes

Reviewer #2: Yes

4. Is the manuscript presented in an intelligible fashion and written in standard English?

Reviewer #1: Yes

Reviewer #2: No

5. Review Comments to the Author

Reviewer #1: The manuscript entitled “Transmission mode of watermelon silver mottle virus by Thrips palmi” improves our understanding on the transmission mode of a Tospovirus of global importance. It is well written and it provides relevant information on the ecology of the disease triangle (watermelon- WSMoV-Thrips palmi).

However, I have two major concerns. The first one relates to data analysis. There is a lack of information on the statistical methods used. Only when looking at the results, authors report that they performed logistic regression (253-255). This should be further explained in the materials and methods section, describing the modelling approach, factors taken into consideration, variables and the statistical package used. Also, Table 1 would benefit from reporting sample sizes (I guess from the methods that authors used n=30 per assay, but it is useful to see it already in the table, especially if some replicates were lost).

My second major point relates to the status of T. palmi as the only vector of the virus. References 5) and 6) state that this species transmits WSMoV, but, to my knowledge, there are no studies looking at the range of potential vectors that could transmit the virus. In this sense, the argument in lines 64-66 may be misleading. In these sentences, it seems like F. occidentalis is not a potential vector of WSMoV, while this may well be, giving the ability of this thrips species to transmit a wide range of tospoviruses. Additionally (Chiemsombat et al. Arch Virol (2008) 153:571–577 DOI: 10.1007/s00705-007-0024-3 ) suggested that another thrips species (Scritothrips dorsalis) could potentially vector the virus. I would encourage authors to further explore this aspect in their text.

In lines 335-336 authors suggest that there is a disagreement between their findings and those of Chen et al. 2006. I would recommend to expand the discussion on mechanical transmission of WSMoV, for example by addressing the following question: What are the different methodologies and findings between the current and previously published work on transmission mechanisms (Chen et al. 2006)?

Few minor points:

-line 98: 150 mesh: please, provide the number of threads/cm2 or the area of the opening.

-line 184: Potential viruliferous or “potentially viruliferous”?

-line 227: In the era of Open Access, are authors going to deposit their sequences in the public database of ncbi?

-line 332: Additionally, Frankliniella fusca is a major vector of TSWV in peanuts, for example.

Reviewer #2: The authors aimed to empirically determine the transmission mode of WSMoV, an orthotospovirus, by Thrips palmi, the reported thrips vector of this plant virus. The type member of the genus orthotospovirus (tomato spotted wilt virus) is transmitted by other thrips vectors in a persistent - propagative manner, meaning that the virus replicates in the vector and disseminates from the midgut to the salivary glands prior to inoculation of plants. As such, the authors performed various transmission experiments with various parameters to distinguish some transmission modes, along with confocal laser scanning microscopy of immunofluorescent detection of WSMoV nucleocapsid (N) protein (structural protein) to document the presence of the viral protein in adult thrips tissues known to be associated with orthotospovirus infection.

Major Concerns:

1. The authors rationalize that endpoint RT-PCR detection of complementary RNA of the NSs or NSm genes is evidence of replication of WSMoV. While it is correct that amplification of the viral complement (vc) of NSs (S genome segment) or NSm (M genome segment) is suggestive of replication (these viral proteins are translated from subgenomic virus sense mRNA), it is not conclusive of replication because it has been reported in foundational studies that the vc of M and/or S RNA segments of TSWV and other orthotospoviruses (e.g., INSV) can be encapsidated in the virion [VIROLOGY 188, 732-741 (1992) (Law et al); Journal of General Virology (1992), 73, 687-693 (Kormelink et al)]. As such, to be rigorous, it is strongly recommended that the authors perform additional experiments to definitively show replication of WSMoV in Thrips palmi - such as evidence of NSs or NSm protein in the thrips vector using western blot analysis or immunolocalization in tissues with antibodies to these nonstructural viral proteins, because to produce these nonstructural proteins, the S and M RNAs must be replicated into full length vc strands followed by mRNA synthesis. The authors chose to use antibodies to the N protein (structural protein) in their confocal microscopy observation of virus in thrips tissues, so they clearly have the skills to dissect and fix thrips tissues and to perform and analyze confocal microscope images. It is recommended they perform these tissue dissections and immunofluorescence with antibodies to NSs or NSm. Also, use of real-time quantitative RT-PCR of NSs and NSm to estimate virus accumulation (increase) over time after removal of viral inoculum may also support their conclusions.

2. The introduction is missing key elements to help the reader understand the logic and rationale of the objective/subobjectives of the study. There is no mention of the genome organization of WSMoV or other orthotospoviruses, which would help readers understand the authors' approach. WSMoV genome is deposited in NCBI Genbank. There is no clear description of possible transmission modes - nonpersistent, semipersistent, persistent non-propagative, persistent-propagative - that set the stage for the acquisition-inoculation experiments. Along these lines, a description of factors that vector biologists use to distinguish transmission modes (i.e., time of virus exposure (AAP), time to virus delivery (IAP), latent period, frequency of inoculation, accumulationo of virions and/or nonstructural proteins). It is unclear to the reader how the authors differentiate or identify the different modes of transmission, and therefore the manuscript would benefit from a clear description in the introduction. And, on lines 87 - 90, there is mention of phylogenetic analysis of N aa sequences between WSMoV and TSWV, but there is no clear logic as to why this is mentioned or relevant to the study.

3. Lines 78 - 86. The assertion that published studies make assumptions about orthotospovirus transmission biology based on TSWV is not fairly supported. In fact, several of the references cited by the authors on line 82 have examined transmission and accumulation of viral protein (nonstructural) of other tospoviruses and different thrips species. It is strongly suggested that this assertion and text be modified to accurately reflect the knowledge of transmission modes of orthotospoviruses and to temper the language as to not offend experts in the field. Along this line, the authors missed a recent study that documented vector competence of another orthotospovirus, soybean vein necrosis virus (SVNV) by soybean thrips (Neohydatothrips variabilis) that carried out the appropriate steps (testing different AAPs and IAPs and confocal microscopy to trace tissue tropism in the vector) to determine transmission - no assumptions were made.

In addition, it seems odd that members of this author group published unrelated research with this vector (T. palmi) and orthotospovirus species (WSMoV) and appeared to work out the AAP and IAP for their experiments (https://doi.org/10.1371/journal.pone.0102021). They produced WSMoV-infected plants under the presumption that L1s acquire the virus (24 hour AAP) and adults transmit (IAP), so it would appear that in this previous paper, the authors presumed that WSMoV is transmitted in the same manner as a persistent virus like TSWV (virus acquired as young L1s, virus retained after molting and pupation, adults inoculate plants).

4. Lack of statistical analysis section for handling transmission data. The authors mention 'logistic regression' (line 255) of the transmission efficiency data in Table 1, but how did the authors handle the three independent experiments, were data pooled, what was the statistical model tested, were assumptions met? Please devote a few lines to describing the value of the 'Swallow estimator' and why it was used here.

Minor comment: The writing could be improved throughout; there were several awkward clauses and ambiguous language.

6. PLOS authors have the option to publish the peer review history of their article (what does this mean?). If published, this will include your full peer review and any attached files.

Reviewer #1: No

Reviewer #2: No

---

## [Author Response · Author response to Decision Letter 0]

24 Dec 2020

Response to Reviewers

Reviewer #1: The manuscript entitled “Transmission mode of watermelon silver mottle virus by Thrips palmi” improves our understanding on the transmission mode of a Tospovirus of global importance. It is well written and it provides relevant information on the ecology of the disease triangle (watermelon- WSMoV-Thrips palmi).

However, I have two major concerns. The first one relates to data analysis. There is a lack of information on the statistical methods used. Only when looking at the results, authors report that they performed logistic regression (253-255). This should be further explained in the materials and methods section, describing the modelling approach, factors taken into consideration, variables and the statistical package used. Also, Table 1 would benefit from reporting sample sizes (I guess from the methods that authors used n=30 per assay, but it is useful to see it already in the table, especially if some replicates were lost).

Response: We consulted a statistician, and the data were analyzed again. The information of statistical analyses was explained in the materials and methods section. Sample sizes were added in Table 1 as suggested.

My second major point relates to the status of T. palmi as the only vector of the virus. References 5) and 6) state that this species transmits WSMoV, but, to my knowledge, there are no studies looking at the range of potential vectors that could transmit the virus. In this sense, the argument in lines 64-66 may be misleading. In these sentences, it seems like F. occidentalis is not a potential vector of WSMoV, while this may well be, giving the ability of this thrips species to transmit a wide range of tospoviruses. Additionally (Chiemsombat et al. Arch Virol (2008) 153:571–577 DOI: 10.1007/s00705-007-0024-3) suggested that another thrips species (Scritothrips dorsalis) could potentially vector the virus. I would encourage authors to further explore this aspect in their text.

Response: There are no published researches studying the range of potential vectors of WSMoV, so we have rewritten the sentences. Also, there are no studies investigating vector competence of F. occidentalis to transmit WSMoV. Chiemsombat et al. (2008) only mentioned that “the ability of these two thrips species to transmit different tospoviruses in Thailand needs further investigation,” but no experiments were performed; therefore, we decide not to cite the paper.

In lines 335-336 authors suggest that there is a disagreement between their findings and those of Chen et al. 2006. I would recommend to expand the discussion on mechanical transmission of WSMoV, for example by addressing the following question: What are the different methodologies and findings between the current and previously published work on transmission mechanisms (Chen et al. 2006)?

Response: Chen et al. (2006) proposed that T. palmi transmits WSMoV both non-persistently and persistently; however, it is not in line with our understanding of virus transmission mode. We evaluated the methods of their experiments and found they were okay. Therefore, we decided to conduct a series of experiments to examine whether T. palmi transmits WSMoV non-persistently, semi-persistently, or persistently. Finally, our results clarify that T. palmi transmits WSMoV in a persistent-propagative mode, and not in a non-persistent or semi-persistent manner.

Few minor points:

 -line 98: 150 mesh: please, provide the number of threads/cm2 or the area of the opening.

Response: The information (pore size 160 μm) was added as suggested.

-line 184: Potential viruliferous or “potentially viruliferous”?

Response: It was corrected.

-line 227: In the era of Open Access, are authors going to deposit their sequences in the public database of ncbi?

Response: The sequences of partial NSm and NSs genes were 100% identical to WSMoV isolate DD6 the M and S segments in GenBank (accession numbers DQ157768 and AY864852, respectively). The information was added in the text.

-line 332: Additionally, Frankliniella fusca is a major vector of TSWV in peanuts, for example.

Response: Thank you for providing the reference. We have added the information and cited the paper (Sakimura, 1963. Phytopathology).

Reviewer #2: The authors aimed to empirically determine the transmission mode of WSMoV, an orthotospovirus, by Thrips palmi, the reported thrips vector of this plant virus. The type member of the genus orthotospovirus (tomato spotted wilt virus) is transmitted by other thrips vectors in a persistent - propagative manner, meaning that the virus replicates in the vector and disseminates from the midgut to the salivary glands prior to inoculation of plants. As such, the authors performed various transmission experiments with various parameters to distinguish some transmission modes, along with confocal laser scanning microscopy of immunofluorescent detection of WSMoV nucleocapsid (N) protein (structural protein) to document the presence of the viral protein in adult thrips tissues known to be associated with orthotospovirus infection.

Major Concerns:

1. The authors rationalize that endpoint RT-PCR detection of complementary RNA of the NSs or NSm genes is evidence of replication of WSMoV. While it is correct that amplification of the viral complement (vc) of NSs (S genome segment) or NSm (M genome segment) is suggestive of replication (these viral proteins are translated from subgenomic virus sense mRNA), it is not conclusive of replication because it has been reported in foundational studies that the vc of M and/or S RNA segments of TSWV and other orthotospoviruses (e.g., INSV) can be encapsidated in the virion [VIROLOGY 188, 732-741 (1992) (Law et al); Journal of General Virology (1992), 73, 687-693 (Kormelink et al)]. As such, to be rigorous, it is strongly recommended that the authors perform additional experiments to definitively show replication of WSMoV in Thrips palmi - such as evidence of NSs or NSm protein in the thrips vector using western blot analysis or immunolocalization in tissues with antibodies to these nonstructural viral proteins, because to produce these nonstructural proteins, the S and M RNAs must be replicated into full length vc strands followed by mRNA synthesis. The authors chose to use antibodies to the N protein (structural protein) in their confocal microscopy observation of virus in thrips tissues, so they clearly have the skills to dissect and fix thrips tissues and to perform and analyze confocal microscope images. It is recommended they perform these tissue dissections and immunofluorescence with antibodies to NSs or NSm. Also, use of real-time quantitative RT-PCR of NSs and NSm to estimate virus accumulation (increase) over time after removal of viral inoculum may also support their conclusions.

Response: We agree with the reviewer’s comment that the detection of complementary RNA of the NSs and NSm genes is not conclusive of virus replication. Therefore, we performed western blot analysis of NSs protein in viruliferous thrips to demonstrate the replication of WSMoV in Thrips palmi (Fig. 4).

2. The introduction is missing key elements to help the reader understand the logic and rationale of the objective/subobjectives of the study. There is no mention of the genome organization of WSMoV or other orthotospoviruses, which would help readers understand the authors' approach. WSMoV genome is deposited in NCBI Genbank. There is no clear description of possible transmission modes - nonpersistent, semipersistent, persistent non-propagative, persistent-propagative - that set the stage for the acquisition-inoculation experiments. Along these lines, a description of factors that vector biologists use to distinguish transmission modes (i.e., time of virus exposure (AAP), time to virus delivery (IAP), latent period, frequency of inoculation, accumulation of virions and/or nonstructural proteins). It is unclear to the reader how the authors differentiate or identify the different modes of transmission, and therefore the manuscript would benefit from a clear description in the introduction. And, on lines 87 - 90, there is mention of phylogenetic analysis of N aa sequences between WSMoV and TSWV, but there is no clear logic as to why this is mentioned or relevant to the study.

Response: The background information of transmission mode of insect transmitted plant viruses was added in the introduction section. The phylogenetic analysis of amino acid sequences of WSMoV and TSWV N proteins demonstrates that WSMoV is distantly related to TSWV. Our objective was to examine the transmission mode of a virus that is distantly related to well-studied prototype, TSWV. We have rewritten the sentences. To avoid an overly long introduction, we did not add the description of WSMoV genome organization because the genome organization of WSMoV is the same as that of other orthotospoviruses.

3. Lines 78 - 86. The assertion that published studies make assumptions about orthotospovirus transmission biology based on TSWV is not fairly supported. In fact, several of the references cited by the authors on line 82 have examined transmission and accumulation of viral protein (nonstructural) of other tospoviruses and different thrips species. It is strongly suggested that this assertion and text be modified to accurately reflect the knowledge of transmission modes of orthotospoviruses and to temper the language as to not offend experts in the field. Along this line, the authors missed a recent study that documented vector competence of another orthotospovirus, soybean vein necrosis virus (SVNV) by soybean thrips (Neohydatothrips variabilis) that carried out the appropriate steps (testing different AAPs and IAPs and confocal microscopy to trace tissue tropism in the vector) to determine transmission - no assumptions were made.

Response: We thank the reviewer for the kind reminding. We have no intention to offend experts in this field. The paragraph addresses the fact that these studies, transmission mode of tospoviruses other than TSWV, only examined the possibility of persistent transmission but not the possibility of non-persistent and semi-persistent transmission modes. We did not mention that these studies did not demonstrate the accumulation of viral nonstructural proteins and virus replication. We have rewritten the sentences to make it more clear. We apologize for missing the recent study of SVNV (Han et al. 2019. Front. Microbiol.), and the paper was cited. However, Han et al. (2019) let first-instar larvae to acquire the virus and inoculated soybean leaf disk with adult thrips. They also did not examine the possibility of non-persistent and semi-persistent transmission modes for SVNV.

In addition, it seems odd that members of this author group published unrelated research with this vector (T. palmi) and orthotospovirus species (WSMoV) and appeared to work out the AAP and IAP for their experiments (https://doi.org/10.1371/journal.pone.0102021). They produced WSMoV-infected plants under the presumption that L1s acquire the virus (24 hour AAP) and adults transmit (IAP), so it would appear that in this previous paper, the authors presumed that WSMoV is transmitted in the same manner as a persistent virus like TSWV (virus acquired as young L1s, virus retained after molting and pupation, adults inoculate plants).

Response: That's right. We presumed that vector transmission of WSMoV is the same as that of TSWV in our previous published paper. Examined thrips acquired the virus at L1 stage and transmitted the virus after emergence as adults. We thought that the transmission of WSMoV needs to be examined in details, and this was our motivation for this research.

4. Lack of statistical analysis section for handling transmission data. The authors mention 'logistic regression' (line 255) of the transmission efficiency data in Table 1, but how did the authors handle the three independent experiments, were data pooled, what was the statistical model tested, were assumptions met? Please devote a few lines to describing the value of the 'Swallow estimator' and why it was used here.

Response: We consulted a statistician, and the data were analyzed again. The information of statistical analyses was explained in the materials and methods section. Swallow estimator was used to estimate the probability that a single thrips transmits the virus when groups of thrips were used for virus inoculation. The information was moved to Statistical analyses section in the materials and methods.

Minor comment: The writing could be improved throughout; there were several awkward clauses and ambiguous language.

Response: English is not our native language, so the manuscript has been edited by a native speaker, Ms. Reina L. Tong, in the submitted version. The revised manuscript has been carefully revised by a professional language editing service to improve the grammar and readability.

---

## [Decision Letter · Decision Letter 1]

2 Feb 2021

PONE-D-20-21623R1

Transmission mode of watermelon silver mottle virus by Thrips palmi

PLOS ONE

Dear Dr. Tsai,

Thank you for submitting your manuscript to PLOS ONE. After careful consideration, we feel that it has merit but does not fully meet PLOS ONE’s publication criteria as it currently stands. Therefore, we invite you to submit a revised version of the manuscript that addresses the points raised during the review process.

Congratulations for this new version of this manuscript, but before acceptance I would like to ask you to please address the minor details suggested by the reviewers. 

We look forward to receiving your revised manuscript.

Kind regards,

Livia Maria Silva Ataide

Academic Editor

PLOS ONE

Reviewers' comments:

Reviewer's Responses to Questions

**Comments to the Author**

1. If the authors have adequately addressed your comments raised in a previous round of review and you feel that this manuscript is now acceptable for publication, you may indicate that here to bypass the “Comments to the Author” section, enter your conflict of interest statement in the “Confidential to Editor” section, and submit your "Accept" recommendation.

Reviewer #1: (No Response)

Reviewer #2: (No Response)

2. Is the manuscript technically sound, and do the data support the conclusions?

Reviewer #1: Yes

Reviewer #2: Yes

3. Has the statistical analysis been performed appropriately and rigorously? 

Reviewer #1: Yes

Reviewer #2: Yes

4. Have the authors made all data underlying the findings in their manuscript fully available?

Reviewer #1: Yes

Reviewer #2: No

5. Is the manuscript presented in an intelligible fashion and written in standard English?

Reviewer #1: Yes

Reviewer #2: Yes

6. Review Comments to the Author

Reviewer #1: I would like to acknowledge that authors addressed well my previous comments to the manuscript entitled “Transmission mode of watermelon silver mottle virus by Thrips palmi”. I have only two considerations in relation to the section of statistics and table 1.

Authors use reference number 38 (Chen et al. 2006) to refer to the calculations of the Swallow estimator (line 266) However, based on the list provided, it is likely that the reference that correspond to the definition of this estimator is number 37 (Swallow, 1985). Therefore, I would suggest to correct the numbering of the reference list and to double check that all the references are in agreement with the statements in the text.

In addition, in table 1, there is one cell in the column “Ps” that remains empty. This corresponds to thrips that acquired the virus as first instar larvae and transmitted as adults. I would encourage authors to also provide this value to complete the information. Also, the formula of Ps could be added to the figure footnotes so that readers can figure out the meaning of this parameter.

Reviewer #2: The authors responded adequately to my comments in the narrative and in performing the additional experiments to provide more supportive evidence for persistent/propagative (viral replication) transmission of WSMoV by T. palmi. This study provides a thorough examination of this orthotospovirus/thrips vector transmission mode. The revised manuscript is more clearly written and presented logically (the rationale for the study is clearly stated in this version), and it contains sufficient background and description of methodology to evaluate the significance of the study

Below are a few minor comments to address in the final revision:

Figures and figure legends.

Figures should stand alone with the use of legends, meaning, the reader should not have to revisit the paper methodologies to fully evaluate and comprehend the content of the figure.

Thus figure legends should include pertinent information:

In both figure 1 and 2 legends, there is no mention of the viral protein detected by the antibody: N or Nsm or Nss? The reader should know this from the legends.

Also, in Figure 2, are the five panels in each column showing a different independent sample, eg. A, E, I, M and Q are internal organs of five different individuals from how many of the repeated experiments? Or a different part of the same tissue of one individual in each column?

Since the authors labeled each panel with letters, they must be meaningful so they should be referred to in the figure legend. Also, in the Results section where Figure 2 is reported, the reader should be told how many samples were tested and if what's depicted in Figure 2 is a representative sample.

Statistical methods section.

Statistics - data was pooled from three biological repetitions of the transmission experiment (Table 1) because authors found that "there were no significant differences of variance among them." (line 292-293). Please include the name of the statistical test performed and provide the test statistic used to assess this. Also, it is increasingly more common to provide supplementary/supportive materials with manuscripts for publication. As such, it is recommended that an excel file or word table be provided that shows % transmission data for each of the three biological repetitions to show repeatability/robustness of findings and thoroughness of the study.

7. PLOS authors have the option to publish the peer review history of their article (what does this mean?). If published, this will include your full peer review and any attached files.

Reviewer #1: No

Reviewer #2: No

---

## [Author Response · Author response to Decision Letter 1]

6 Feb 2021

Response to Reviewers

Reviewer #1: I would like to acknowledge that authors addressed well my previous comments to the manuscript entitled “Transmission mode of watermelon silver mottle virus by Thrips palmi”. I have only two considerations in relation to the section of statistics and table 1.

Authors use reference number 38 (Chen et al. 2006) to refer to the calculations of the Swallow estimator (line 266) However, based on the list provided, it is likely that the reference that correspond to the definition of this estimator is number 37 (Swallow, 1985). Therefore, I would suggest to correct the numbering of the reference list and to double check that all the references are in agreement with the statements in the text.

Response: The citations in the manuscript are carefully double-checked.

In addition, in table 1, there is one cell in the column “Ps” that remains empty. This corresponds to thrips that acquired the virus as first instar larvae and transmitted as adults. I would encourage authors to also provide this value to complete the information. Also, the formula of Ps could be added to the figure footnotes so that readers can figure out the meaning of this parameter.

Response: Because one thrips per test plant was used for virus inoculation, there is no Swallow estimator for the cell in Table 1. The formula of Ps is added to the footnotes.

Reviewer #2: The authors responded adequately to my comments in the narrative and in performing the additional experiments to provide more supportive evidence for persistent/propagative (viral replication) transmission of WSMoV by T. palmi. This study provides a thorough examination of this orthotospovirus/thrips vector transmission mode. The revised manuscript is more clearly written and presented logically (the rationale for the study is clearly stated in this version), and it contains sufficient background and description of methodology to evaluate the significance of the study

Below are a few minor comments to address in the final revision:

Figures and figure legends.

Figures should stand alone with the use of legends, meaning, the reader should not have to revisit the paper methodologies to fully evaluate and comprehend the content of the figure.

Thus figure legends should include pertinent information:

In both figure 1 and 2 legends, there is no mention of the viral protein detected by the antibody: N or Nsm or Nss? The reader should know this from the legends.

Response: Nucleocapsid protein was detected by the antibody. Information of the antibody is added to Figures 1 and 2 legends.

Also, in Figure 2, are the five panels in each column showing a different independent sample, eg. A, E, I, M and Q are internal organs of five different individuals from how many of the repeated experiments? Or a different part of the same tissue of one individual in each column?

Since the authors labeled each panel with letters, they must be meaningful so they should be referred to in the figure legend. Also, in the Results section where Figure 2 is reported, the reader should be told how many samples were tested and if what's depicted in Figure 2 is a representative sample.

Response: Yes, they are confocal images of organs from five different individuals. Figure 2 is modified, and the figure legend is edited. In the Results section, the information is already described in the manuscript (lines 332-4).

Statistical methods section.

Statistics - data was pooled from three biological repetitions of the transmission experiment (Table 1) because authors found that "there were no significant differences of variance among them." (line 292-293). Please include the name of the statistical test performed and provide the test statistic used to assess this. Also, it is increasingly more common to provide supplementary/supportive materials with manuscripts for publication. As such, it is recommended that an excel file or word table be provided that shows % transmission data for each of the three biological repetitions to show repeatability/robustness of findings and thoroughness of the study.

Response: The statistical test performed to examine variance among 3 replicates is added (lines 304-5 and 316-8). We also add a supplementary table as suggested.

---

## [Editor Report · Decision Letter 2]

9 Feb 2021

Transmission mode of watermelon silver mottle virus by Thrips palmi

PONE-D-20-21623R2

Dear Dr. Tsai,

We’re pleased to inform you that your manuscript has been judged scientifically suitable for publication and will be formally accepted for publication once it meets all outstanding technical requirements.

Kind regards,

Livia Maria Silva Ataide

Academic Editor

PLOS ONE
---

## [Editor Report · Acceptance letter]

22 Feb 2021

PONE-D-20-21623R2 

Transmission mode of watermelon silver mottle virus by *Thrips palmi*

Dear Dr. Tsai:

I'm pleased to inform you that your manuscript has been deemed suitable for publication in PLOS ONE. Congratulations! Your manuscript is now with our production department. 

Kind regards, 

on behalf of

Dr. Livia Maria Silva Ataide 

Academic Editor

PLOS ONE